# Mini-Review: Current Bladder Cancer Treatment—The Need for Improvement

**DOI:** 10.3390/ijms25031557

**Published:** 2024-01-26

**Authors:** Emily Gill, Claire M. Perks

**Affiliations:** Cancer Endocrinology Group, Learning & Research Building, Southmead Hospital, Translational Health Sciences, Bristol Medical School, Bristol BS10 5NB, UK; vw19109@bristol.ac.uk

**Keywords:** bladder cancer treatment, non-muscle-invasive bladder cancer, muscle-invasive bladder cancer, radial cystectomy, transurethral resection of the bladder tumour, biomarkers

## Abstract

Bladder cancer is the tenth most common cancer and is a significant burden on health care services worldwide, as it is one of the most costly cancers to treat per patient. This expense is due to the extensive treatment and follow-ups that occur with costly and invasive procedures. Improvement in both treatment options and the quality of life these interventions offer has not progressed at the rates of other cancers, and new alternatives are desperately needed to ease the burden. A more modern approach needs to be taken, with urinary biomarkers being a positive step in making treatments more patient-friendly, but there is still a long way to go to make these widely available and of a comparable standard to the current treatment options. New targets to hit the major signalling pathways that are upregulated in bladder cancer, such as the PI3K/AkT/mTOR pathway, are urgently needed, with only one drug approved so far, Erdafitinib. Immune checkpoint inhibitors also hold promise, with both PD-1 and CDLA-4 antibody therapies approved for use. They effectively block ligand/receptor binding to block the immune checkpoint used by tumour cells. Other avenues must be explored, including drug repurposing and novel biomarkers, which have revolutionised this area in other cancers.

## 1. Introduction

Bladder cancer (BCa) is the tenth most common cancer worldwide, although the incidence rate is three to four times higher in men than women. BCa is generally considered an age-related disease, as many of the cases appear in those over the age of 75 [1]. Whilst incidence rates of BCa in women are lower than those seen in men, women generally develop a more aggressive form of the disease and present at a more advanced stage than their male counterparts. Due to this, women also tend to have a worse outcome. There are many possible reasons for this, although a delay in diagnosis due to symptoms being related to menstruation or cystitis is thought to be a significant cause of the variance between the sexes [2]. Sex hormones are another obvious reason for this variation, with androgens shown to increase the risk of BCa initiation and oestrogens having a protective effect. However, with the decrease in sex hormone production with increasing age, this cannot be the only reason for the sex differences seen in BCa [3].

Risk factors for BCa, which can be seen in Figure 1, largely revolve around exposure to a range of carcinogens, with tobacco smoke accounting for approximately 50% of BCa cases [4]. Whilst vaping is considered a better alternative to traditional cigarettes, the substances that are significantly linked with BCa have been found in the urine of both vape users and tobacco smokers [5]. Occupational exposures, particularly to aromatic amines, are also an important risk factor, with approximately 10% of BCa cases being attributed to working environments. In less developed countries, particularly in parts of Africa, schistosomiasis infection is a considerable risk factor for BCa [6].

Approximately 95% of BCa cases are urothelial cell carcinoma (UCC), the most common histological subtypes of which are conventional urothelial carcinoma, micropapillary carcinoma, and squamous differentiation [7]. It is known that histological subtyping can help predict the prognosis in BCa, with different histological phenotypes observed in those presenting at an advanced stage or with worse predicted outcomes [8]. It is important that these histological variants are identified and reported after the patient has undergone a biopsy, as this could help improve the treatment options at an earlier stage [9].

As seen in breast cancer, it is well accepted that bladder tumours also have distinct molecular subtypes, alongside histological variants that can have an important impact on the tumour behaviour, although the exact number of subtypes is still unclear. Two subtypes are widely accepted, with it being generally acknowledged that other subtypes exist. The two most common subtypes are basal and luminal, as seen in other cancers, but it is often considered that these are umbrella terms, with more specific subgroups being defined within each of these classifications [10]. Novel subtypes are being put forward as research progresses, largely focusing on the up- or downregulation of specific or a set of genes. For example, Cai et al. suggests a metabolic subtype characterised by an increased expression of S100A7 [11]. Unfortunately, the routine use of histological and molecular subtypes to direct therapy with any specificity is currently limited due to the heterogeneity of tumours, crossovers being made between the different subtypes, and whether there is a need for more distinctive groups to be defined [7,12,13,14]. BCa can often present with multiple tumours within the bladder, which can have differing molecular subtypes. In addition, the subtype of the metastatic tumour is not necessarily the same as the primary tumour. This was shown in a study by Sjödahl et al., where patients with basal primary tumours often had metastases with luminal properties, whereas primary tumours with a luminal molecular subtype were more likely to retain that subtype in metastatic tumours [10,15].

There are two distinct groups of BCa, as shown in Figure 2, non-muscle-invasive (NMIBC) and muscle-invasive bladder cancer (MIBC), which represent different stages of the disease. NMIBC often protrudes into the bladder void and includes tumours that enter the connective tissue lying beneath the endothelium. Carcinoma in situ (CIS) is another form of NMIBC that grows flat along the bladder endothelium and is thought to be a precursor to a more aggressive disease than traditional NMIBC. MIBC grows through the muscle layers of the bladder wall before metastasizing to local and distant lymph nodes and organs [1].

Approximately 70% of new bladder cancer cases are grouped into NMIBC stages, with 20% of those going on to develop MIBC. Furthermore, 20% of cases present with MIBC, with 50% of those later developing metastasis despite aggressive treatment [16]. Whilst the 5-year survival rate for those with NMIBC is approximately 90%, long-term monitoring and repeated interventions lead to BCa being one of the least cost-effective cancers in terms of treatment, and it reduces the quality of life of patients for the rest of their lifespan [6].

If bladder cancer is suspected, patients undergo a cystoscopy to confirm the presence of a tumour. This is where a small camera is passed through the urethra to enable the bladder lining to be examined. Recent developments in this area have led to two enhanced methods of cystoscopy: narrow-band imaging, and blue-light cystoscopy. Blue-light cystoscopy has increased the number of tumours being detected versus conventional cystoscopy, whereas narrow-band imaging also has an improved detection rate compared to conventional cystoscopy; it has reduced the number of cases that present with recurrence within 12 months after treatment. Whilst imaging is not commonly used in the diagnosis of BCa, it can be used to aid in the staging of MIBC [17].

## 2. Current Treatment Options

Once BCa has been diagnosed, the treatment pathways are very different for NMIBC versus MIBC. In NMIBC, a transurethral resection of the bladder tumour (TURBT) is performed to remove the tumour, which involves a thin, long instrument being inserted into the bladder with a light, camera, and cutting implement attached. This is similar to a cystoscopy, where a similar instrument is used but without the cutting tool. Patients often require multiple TURBT procedures depending on several factors, including whether all the tumour was removed and the stage and grade of the tumour. Patients are then monitored using cystoscopies. Those tumours that are most at risk of progressing to MIBC will receive Bacille Calmette–Guérin (BCG) treatment for up to 3 years, depending on their response and tolerance, whereas those with lower-grade tumours are more likely to be offered a dose of mitomycin C, a chemotherapeutic agent to reduce any circulating tumour cells.

BCG is an attenuated form of *Mycobacterium bovis*; the specific mechanism of this treatment is not fully understood, but it is thought that the introduction of BCG to the tumour cells leads to the internalisation of BCG. This in turn increases the antigen-presenting complexes on the cell surface, inducing a cytokine-based immune response. This immune cascade increases the activity of the immune cells to detect and destroy the tumour cells. The treatment periods are six weeks in length, with one course given weekly [17,18].

MIBC is currently treated with a radical cystectomy and pelvic lymph node dissection alongside neoadjuvant chemotherapy treatment with a cisplatin-based combination of chemotherapies. There are also options to retain the bladder, such as partial cystectomy and chemoradiation, but these are only viable in selective cases. A radical cystectomy is the removal of the bladder along with local tissues; in men, this includes the prostate and seminal vesicles, whereas in women, the uterus, fallopian tubes, ovaries, and anterior vagina are removed. Lymph node dissection is often recommended at the same time, as this reduces the chance of further metastasis [17].

## 3. Limitations with Current Treatments

One of the most pressing limitations with the current bladder cancer treatment options is the quality of life for the patients. Treatments for many other cancers have moved forward, and improvements have been made to ensure patients have the best possible options, with targeted therapies, such as hormone treatments and scans, rather than invasive methods becoming the norm [19]. However, BCa has been left behind in this respect. On top of this, BCa is one of the most costly to treat overall and is the most expensive cancer to treat per patient, costing between USD 96,000 and USD 187,000 due to the need for regular treatments and follow-ups, which is a strain on both health services and patients’ ability to maintain a normal routine in their lives [20,21].

Radical cystectomy is the current gold standard for MIBC, but this has a huge impact on a patient’s quality of life, with many suffering from associated surgical complications. Due to the number of tissues removed, particularly in women, many aspects of life are affected by a radical cystectomy. Incontinence and urine retention are both common, with bladder replacements not allowing for the same level of control. A loss of sexual function can also have an impact on patients, particularly in those who are diagnosed at a younger age [22]. Studies looking at the quality of life of bladder cancer patients have highlighted that the toll on mental health persists well after the initial treatment, as does the decline in general health [23]. Linked to this, there is an increase in suicide risk amongst those patients that have undergone a radial cystectomy, particularly in those older patients that have developed metastasis [24].

High recurrence rates are seen with TURBT in NMIBC, and a proportion of this can possibly be attributed to the limitations of the treatment. Whilst the best efforts are made, it can be difficult to ensure complete removal of the tumour, and there is a risk of tumour re-implantation following the procedure [25]. Day-case procedures are also being trialled for TURBT to reduce the length of hospital stays. This was well received by patients, with only a small number of patients failing to be discharged on the same day and only 11% of patients discharged as day cases having to be re-admitted due to surgical complications [26].

Whilst most patients tolerate BCG treatment well, many experience side effects. Over 80% of patients reported bladder-related symptoms, including cystitis, dysuria, and haematuria. A significant number of patients stop treatments after the first 6-week cycle and do not continue with the maintenance doses of BCG, with some studies reporting that less than 20% of patients complete the full course of maintenance BCG treatments. Most of the adverse effects reported are considered minor and not life-threatening; however, a small number of BCG-related deaths have been reported following BCG therapy due to sepsis or disseminated mycobacteriosis. Systemic adverse effects are also common, with fever and BCGitis reported in up to half of cases, but more serious side effects such as meningitis or Guillain–Barré syndrome are very rarely reported [18,27,28].

## 4. Urinary Biomarkers

Biomarkers are a significant area of research and are being used in clinical settings to help identify and monitor a large number of diseases. The World Health Organization has stipulated that the definition of a biomarker, or a biological marker, is “almost any measurement reflecting an interaction between a biological system and a potential hazard, which may be chemical, physical, or biological. The measured response may be functional and physiological, biochemical at the cellular level, or a molecular interaction” [29]. One example of this is prostate-specific antigen (PSA). PSA is a prostate-specific enzyme produced by the prostate gland. When measured in the blood, the serum concentration of PSA is generally low. However, when a tumour is present, the serum PSA often significantly increases. This has led to PSA screening to aid in the diagnosis of prostate cancer [30]. For BCa, unsurprisingly, there is a lot of interest around urinary biomarkers due to urine being in direct contact with the tumour(s) in the bladder. A significant amount of research is being carried out in this area, with six tests approved by the Food and Drug Administration (FDA) in various capacities, which can be found in Table 1. Despite this, cytology is currently the only urine-based diagnostic tool that is recommended, although this is suggested for use alongside cystoscopy due to the reduced sensitivity of this technique in detecting low-grade BCa [31].

NMP22 is a urinary biomarker for use in both surveillance and the diagnosis of BCa cases. Whilst both NMP22 tests have been approved for the surveillance of patients, only BladderChek^®^ has been approved for diagnostic use in those patients deemed to be at higher risk of developing BCa or patients that are displaying symptoms of BCa. The hCFHrp urinary biomarker tests shown in Table 1 are only approved for use in detecting recurrence alongside a cystoscopy in the monitoring of previously treated patients. This is likely due to the large number of false positives received from the low specificity of these tests [32,33].

UroVysion^®^ has been authorised to diagnose and monitor BCa cases, but to use this test, strict conditions must be met. When used, this test has high specificity and sensitivity, particularly with high-grade tumours or those patients with carcinoma in situ, as it is unaffected by conditions that can lead to false positives in some of the other tests mentioned here. ImmunoCyt^®^ has been approved to be used alongside the current methods to improve the diagnosis and monitoring of BCa patients. This method also has a significantly reduced number of false positives compared to the majority of the other approved tests listed in Table 1 [32,34].

NMP22 is a non-histone chromatic protein located in the nucleus which ensures the correct positioning of chromatids during mitosis and the detachment of daughter cells. It is also involved in the nuclear structure, as well as playing a role in DNA replication, RNA synthesis, and transcription. NMP22 is released from dying urothelial cells, so low levels can be detected in healthy individuals, with significantly increased levels seen in BCa patients [35,36].

hCFHrp is produced by Bca tumour cells, but it is not detected in healthy bladder tissue and is thought to be involved in allowing the tumour cells to avoid detection by the immune system by downregulating the alternative complement pathway (ACP) [37]. The ACP is constantly active, continually monitoring all cell surfaces it encounters. When the pathway identifies damaged or diseased cells, complement component 3 (C3) is cleaved into C3a and C3b, with C3b continuing the cascade of C3 cleavage, which in turn can activate other immune response pathways [38,39].

## 5. Time for a More Modern Approach?

Research on the best way to improve treatment for BCa is ongoing, but little headway has yet been made. Urinary biomarkers are a big area of research with the hope of reducing the number of invasive cystoscopy procedures, whilst other researchers are focusing on biomarkers of survival or treatment outcomes. Urinary biomarkers could help with both the diagnosis and management of BCa; currently, six urinary biomarker tests have been approved by the FDA, which are shown in Table 1. The tests in Table 1 show that some progress is being made in this area, but due to the limitations and specifics of them, further biomarkers need to be discovered or detection methods improved. There are also improvements in surgical options for MIBC with the hope that bladder-sparing techniques can be made more widely available [20,32].

Cross-talk between signalling pathways is commonplace and can underlie resistance to targeted therapies. Several signalling pathways have been highlighted as being important in BCa, and understanding the molecular interactions that exist may provide an insight into novel ways to target these key cancer pathways for therapeutic means.

Unsurprisingly, the phosphoinositide 3-kinase/AkT/mammalian target of rapamycin (PI3K/AkT/mTOR) pathway has a role to play in BCa. For example, the loss or mutation of the tumour suppressor gene, the phosphatase and tensin homolog (PTEN) which is the negative regulator of the PI3K/AkT/mTOR pathway, and the upregulation of the mechanistic target of rapamycin (mTOR) activity have been reported in up to 30% and 55% of MIBC cases (as reviewed in [40]). PTEN suppression has been linked with a poor prognosis in BCa and is associated with the development of chemoresistance. Overall, 25% of BCa cases show a mutation to the PIK3CA gene, which codes for the catalytic domain of PI3K [41]. AkT mutations have also been implicated with the tumour necrosis factor-related apoptosis-inducing ligand (TRAIL) pathway, reducing its ability to induce apoptosis in the tumour cells [42]. In the Ras/Raf/MEK/mitogen-activated protein kinase (MAPK) pathway, a mutation to either or both of Ras and fibroblast growth factor receptor 3 (FGFR3) is seen in 85% of NMIBC cases. These modifications to the PI3K and MAPK pathways promote cell survival, unrestrained proliferation, and resistance to therapy, thereby potentially providing numerous potential drug targets [40,43,44].

The relevance of other signalling pathways in BCa, such as notch and wingless (Wnt)/β-catenin, are somewhat controversial, having been shown to both promote and inhibit bladder cancer progression depending on the context (as reviewed in [45,46]). The Wnt/β-catenin pathway is thought to be dysregulated in over 70% of chemo-resistant MIBC cases [43], thereby playing an important role in treatment efficacy.

Cancer nanomedicine is a rapidly developing field which focuses on the use of nanotechnology to deliver targeted therapy to cancer cells without damaging the surrounding healthy tissue. Several types of nanoparticles have been developed, such as inorganic, polymeric, and lipid-based nanoparticles, which have been designed to deliver chemotherapy, immunotherapy, or nucleic acid-based therapy to tumour cells. These nanoparticles are most commonly used due to their biocompatibility. Clinical trials are ongoing for a number of cancer-based nanomedicines, but challenges remain. Patient selection will be critical due to the heterogeneous nature of tumours, although the effects of using nanomedicines are largely unknown as the majority of work has been performed in vivo [47,48,49]. Perhaps nanomedicine can play an important role in the progression of BCa treatment. A study by Erdogar et al. used cationic chitosan nanoparticles, which are positively charged nanoparticles with mucoadhesive properties, to deliver targeted BCG treatments to reduce the side effects in rats, finding a significant improvement compared to the traditional application methods [50]. Kong et al. also utilised mucoadhesive nanoparticles to deliver KDM6A mRNA into *Kdm6a*-null tumours to increase KDM6A expression. KDM6A is a tumour suppressor gene that is often mutated in BCa. Intravesical therapy was employed by delivering the nanoparticles through a catheter directly into the bladder void in an attempt to ensure the treatment reached the required site and in a high enough volume to be effective. Intravesical therapy is not always an option in BCa, as treatments can be removed from the bladder when voided, reducing the concentration and contact time, but the use of nanoparticles means that this could be possible in the future [51].

## 6. Targeted Drugs and Drug Repurposing

Targeted drugs have shown critical benefit to the treatment plans of other cancers, such as colorectal or breast cancer, such as Trastuzumab, a monoclonal antibody treatment for aggressive HER-2-positive breast cancer [52]. These benefits have not been seen in BCa due to the limited effectiveness of the drugs coupled with the complications they pose to patients. One of the few targeted drugs that has shown acceptable results in BCa patients is Erdafitinib, a fibroblast growth factor receptor (FGFR) 1-4 tyrosine kinase inhibitor (shown in Figure 3), which has been approved for use in metastatic BCa patients with FRGR 2 or FGFR 3 mutations that have previously responded to chemotherapy treatment. However, toxicity to Erdafitinib is a point of concern, and the selection of patients is critical. FGFR inhibitors are not effective in patients carrying certain point mutations of the FGFR gene, although there is some evidence to suggest co-therapy with Src inhibitors can help overcome the resistance caused by these mutations [53,54].

Immunotherapy is another exciting option when looking at alternative treatment options in cancer, with a significant volume of work being undertaken in this field. All five immunotherapy drugs that have been approved by the FDA for use in late-stage BCa are immune checkpoint inhibitors (ICI), with Avelumab being one of the first to be sanctioned in 2020. Immune checkpoints are vitally important pathways for regulating the monitoring of cell health and protecting against autoimmunity. Tumour cells are particularly good at evading this immune mechanism and can avoid detection by the immune checkpoint pathways initiated by T cells. Healthy cells present programmed death-1 ligand 1 (PD-L1) on their cell membranes, which binds to the programmed death-1 (PD-1) receptors on the T cell membrane, which inhibits an immune response through limiting cell growth and cytokine release. This response also occurs with cytotoxic T lymphocyte-associated antigen-4 (CTLA-4). When the CTLA-4 receptor is activated on the T cell membrane, the immune checkpoint response inhibits the initiation of an immune response by blocking the cell cycle of the T cell as well as inhibiting interleukin 2 (IL-2) translation. Tumour cells can mimic the normal response that occurs in healthy cells to avoid detection. Whilst it is encouraging that five ICI’s have been approved for use in late-stage BCa, the efficacy of these drugs, however, is low, at less than 25%. Patient selection is key, and clinical trials are ongoing as to how best to employ them. Co-therapy appears to be the best approach so far, although more work needs to be conducted to identify patients that would benefit the most from this kind of treatment, including those with less severe disease [53,55,56].

Drug targets are lacking in BCa, the identification of which could help improve the effectiveness of current approaches. Drug repurposing could be a useful tool, with research aimed at evaluating the effectiveness of current drugs used for other cancers or diseases, such as metformin, a type 2 diabetes drug that has shown promise in reducing the recurrence and progression of NMIBC and is currently undergoing clinical trials [57]. Another example is tamoxifen, a drug given to oestrogen-receptor-positive breast cancer patients in place of post chemotherapy treatment [58], but tamoxifen has also been shown to increase sensitivity to cisplatin in chemo-resistant MIBC but is yet to be approved for clinical use [59]. This is a method that has been successfully employed in many different medical fields, with some examples shown in Table 2.

## 7. Bladder Sparing Alternatives

Bladder-sparing options are desperately needed to improve the lives of those with MIBC; one example of this is trimodal therapy (TMT), which is currently available but not readily utilised, with traditional approaches favoured. This could largely be due to concerns about metastasis and its being considered less effective in the more advanced stages of BCa. TMT consists of TURBT, chemotherapy, and radiotherapy combined, therefore removing the need for bladder removal [62].

The selection criteria for TMT are reasonably restrictive, but it is the range of risk factors that suggest a poor prognosis following treatment that is also problematic when making the decision about who is an optimal candidate [62]. Patient selection factors include identifying those that can tolerate chemotherapy and radiation to the pelvic region, which comes with its own list of gastrointestinal side effects, as well as those that do not have a history of lower urinary tract conditions. There are also factors that must be considered relating to the tumour itself, such as the tumour volume, the involvement of local tissues, or the presence of hydronephrosis [63]. TMT could be ideal for patients over the age of 80, as radial cystectomy is a less suitable option for the elderly due to the surgical risk. Biomarkers could be the answer to help find patients that would best respond to TMT; for example, alterations in the MRE11 homolog, double-strand break repair nuclease (MRE11), and the excision repair cross-complementation 1 and 2 (ERCC1/2) DNA repair pathways are associated with more chemotherapy- and radiation-sensitive tumours [64].

## 8. Potential Biomarkers

Biomarkers could have a hugely important role to play in the improvement of BCa treatment and surveillance. The clinical implications of each biomarker are specific to the information provided and their specific functions, but biomarkers generally inform on the presence, stage, recurrence, or progression of a disease. Novel biomarkers such as pleckstrin homology domain-containing S1 (PLEKHS1) have been mentioned in the context of cancer progression, with particular interest shown in BCa. PLEKHS1 is a gene located on chromosome 10 which codes for a protein of unknown function. PLEKHS1 can carry a mutation in its promotor region consisting of two single nucleotide substitutions, flanked by palindromic sequences. This mutation is particularly high in BCa, with approximately 40% of cases containing the mutation [65]. Interestingly, PLEKHS1 mutations can be identified in urine samples taken from BCa patients, with suggestions that they could be used as part of a panel to monitor patients to reduce the need for invasive procedures. Its full potential as a biomarker may be realised when a better understanding of its function is determined [34].

Vascular endothelial growth factor C (VEGF-C) is thought to be involved in the development of new lymphatic vessels from those currently existing, a process known as lymphangiogenesis. It is currently inconclusive from the current literature whether VEGF-C levels are significantly altered in BCa cases with lymph node metastasis, but it has repeatedly been shown to be involved in the growth and invasion capabilities of aggressive bladder cells, as well as being linked to chemoresistance in BCa cell lines. VEGF-C could be an ideal candidate marker for assessing chemoresistance in patients prior to treatment and potentially as a marker of progression in the future [66,67,68].

FGFR3 is a relatively well-known biomarker across several cancer types that regularly carries mutations and is involved in a signalling cascade resulting in a wide range of processes within the cell, including differentiation and proliferation. Whilst FGFR3 signalling can be activated in all bladder tumours, regardless of stage or mutation, a single nucleotide substitution is associated with an increased risk of recurrence in NMIBC [69,70].

The three possible future biomarkers for BCa discussed here show the versatility that biomarkers can provide, with potential markers to diagnose, monitor, and predict the course of BCa on a patient-by-patient basis. This kind of approach could have a huge impact on the quality of life for patients, reducing invasive procedures and indicating the best treatment options. However, much more work is required before that stage is achieved, but it is an area well deserving of investment.

## 9. Conclusions

It is clear from other cancers that the ability to develop targeted therapies can culminate in better overall survival and less drug toxicity. The immunotherapy approaches for bladder cancer are promising but are currently not effective enough to be used as a single treatment. Co-therapy is an attractive option, either alongside chemotherapy or perhaps a combined immunotherapy approach. Given the rates of progression and the difficulties BCG treatment poses, perhaps some ICIs could be pre-directed for use at early stages instead of as a last resort. Drugs that target the major signalling pathways need to be further explored, as whilst Erdafitinib has been developed, its toxicity is problematic and needs to be overcome to improve its efficacy. Perhaps focus needs to turn to biomarkers such as PLEKHS1 so in the future BCa can be managed and treated in a less invasive and life-altering way. These biomarkers and treatment options may not be able to remove the need for the current treatments, but they could at least allow some sense of freedom from the regimes and difficulties faced by current BCa patients, which are some of the least invested in and most costly in comparison with those of other cancers.

All the papers were identified using PubMed using our defined keywords and then selected based on their relevance.

## Figures and Tables

**Figure 1 ijms-25-01557-f001:**
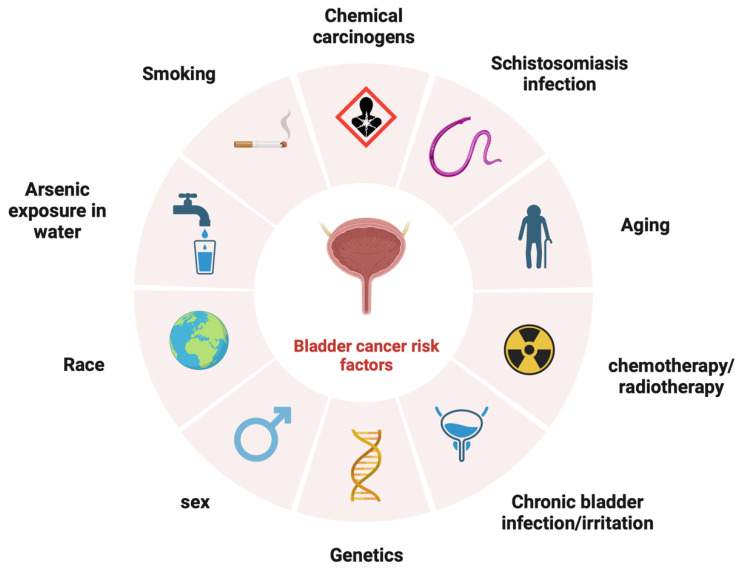
Risk factors for bladder cancer. Created by BioRender.com (accessed on 27 October 2023).

**Figure 2 ijms-25-01557-f002:**
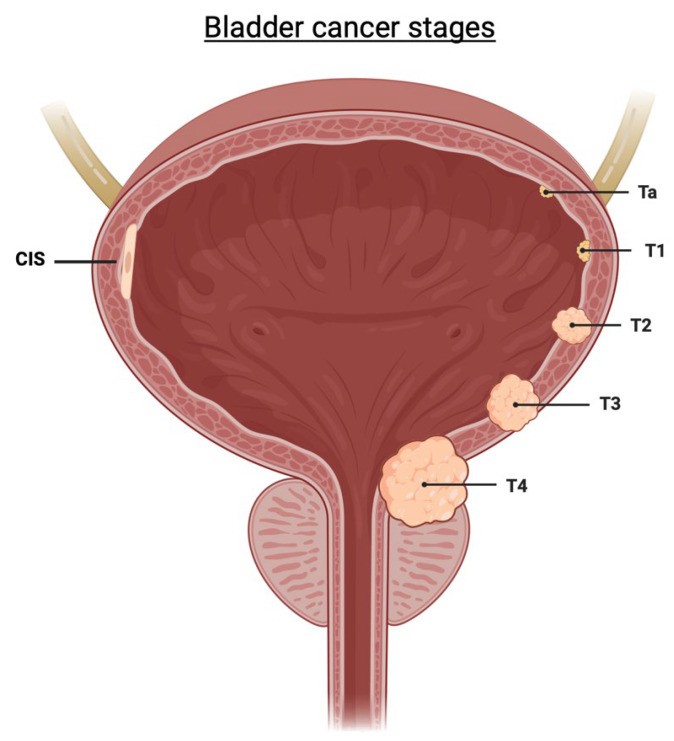
The different stages of bladder cancer: non-muscle-invasive stages consisting of the CIS, Ta, and T1 stages, and muscle invasive stages T2, T3, and T4. Created with BioRender.com (accessed on 27 October 2023).

**Figure 3 ijms-25-01557-f003:**
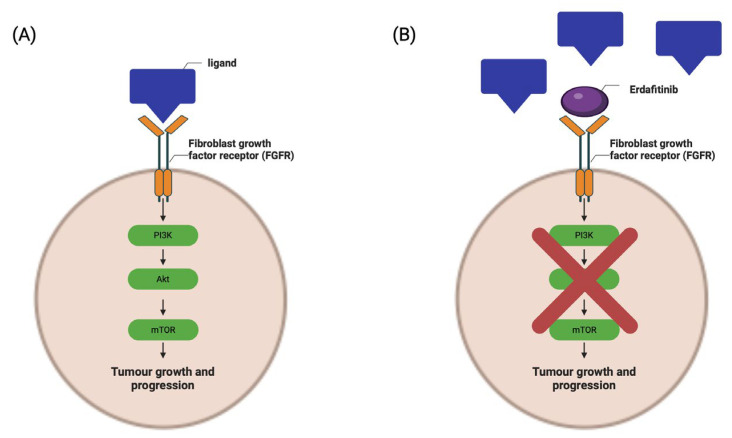
(**A**) The downstream effects of FGFR receptors when activated by a ligand. (**B**) How this signalling pathway is disrupted by Erdafitinib, inhibiting the progression of tumour cells through blocking the activation of the PI3K/AkT/mTOR signalling cascade. Created using BioRender.com (accessed on 30 November 2023).

**Table 1 ijms-25-01557-t001:** FDA-approved urinary biomarker tests for the diagnosis and monitoring of BCa [32].

Test	Biomarker	Availability
BladderChek^®^	Nuclear matrix protein 22 (NMP22)	Point of care test
ImmunoCyt^®^	Carcinoembryonic antigens and sulphated mucin glycoproteins	Laboratory-based immunocytoflourescence test
NMP22^®^ Bladder Cancer Test	Nuclear matrix protein 22 (NMP22)	Laboratory-based ELISA test
BTA Stat	Human complement factor H-related protein (hCFHrp)	Point of care test
BTA TRAK	Human complement factor H-related protein (hCFHrp)	Laboratory-based ELISA test
UroVysion^®^	Aneuploidy of chromosomes 3, 7, or 17 and the loss of the 9p21 locus	Laboratory-based fluorescence in situ hybridisation (FISH) test

**Table 2 ijms-25-01557-t002:** Repurposed drugs with their original and newly established use [60,61].

Drug	Original Use	Repurposed Use
Zidovudine	Cancer	HIV
Minoxidil	Hypertension	Hair growth
Doxorubicin	Anthracycline antibiotic	Cancer
Thalidomide	Sedative	Multiple myeloma
Sildenafil	Hypertension and angina	Erectile dysfunction

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
