# Peer review of "Mini-Review: Current Bladder Cancer Treatment—The Need for Improvement"

_ijms, 2024, doi:10.3390/ijms25031557_

Round 1
Reviewer 1 Report
Comments and Suggestions for Authors
Emily Gill and Claire M. Perks submitted an interesting mini review about bladder cancer therapy. The topic was to some degree significance, and might arouse a certain impact in its field. However, there were some flaws in the current version. Overall, a Major Revision was needed before final acceptance. Please refer to the following comments:
I. Please do not use abbreviations as Keywords, which was unfavorable for understanding.
II. It would be helpful to add some information about the histopathology of BCa in the Introduction.
III. In Section 4, some emerging therapies using nanomedicines should be discussed.
IV. The standing-alone of Section 5 was to some extent illogical. This section might be associated with BCa diagnostics, and could be moved to Section 2.
V. It was insufficient to only demonstrate PLEKHS1 as a new biomarker. The potential implications for clinical application should be stated.
Author Response
We would like to thank the reviewer for taking the time to evaluate this manuscript and for their valued comments and feedback.
1. Please do not use abbreviations as Keywords, which was unfavorable for understanding.
Thank you for this helpful comment. All abbreviations have now been removed from the keywords list.
2. It would be helpful to add some information about the histopathology of BCa in the Introduction.
Thank you for this valuable suggestion, additional information about the histopathology of BCa has been included in from lines 48 – 55.
3. In Section 4, some emerging therapies using nanomedicines should be discussed.
Thank you for this interesting point. A paragraph about the use of nanomedicines has now been included from line 250 – 270.
4. The standing-alone of Section 5 was to some extent illogical. This section might be associated with BCa diagnostics and could be moved to Section 2.
Thank you for this suggestion. We agree that section 5 should be moved, and it has now been placed as section 4 as we feel this is the most logical place following the revisions to the manuscript. However, we would be happy to move to section 2 if the reviewer still feels that this would be the best place.
5. It was insufficient to only demonstrate PLEKHS1 as a new biomarker. The potential implications for clinical application should be stated.
Thank you for this constructive comment. Other biomarkers have now been added and clinical implications discussed in lines 361 – 379.
Reviewer 2 Report
Comments and Suggestions for Authors
This manuscript presents a mini-review on the current progress and possible future directions of bladder cancer treatment
Minor comments:
1. Please keep the font of all the table and figure captions consistent.
2. “Chapter 5. Urinary biomarkers” needs a short paragraph to firstly introduce biomarker technique.
Author Response
We would like to thank the reviewer for taking the time to evaluate this manuscript and for their valued comments and feedback.
This manuscript presents a mini review on the current progress and possible future directions of bladder cancer treatment
Minor comments:
- Please keep the font of all the table and figure captions consistent.
Thank you for this helpful comment. The font of all figure captions is now all the same. As can be seen in line 39, 88-89, 184, 289-292, and 325.
- “Chapter 5. Urinary biomarkers” needs a short paragraph to firstly introduce biomarker technique.
Thank you for this valued comment. A paragraph has been added to introduce biomarkers, as seen on lines 167-183.
Reviewer 3 Report
Comments and Suggestions for Authors
The review is indeed minimalistic. The Authors of the work presented basic information about current bladder cancer treatment, which is particularly important for both parties - patients and doctors.
The methods mentioned in the work are generally known to specialists, but the manner and scope of discussion of currently used bladder cancer treatment methods is significantly limited for non-specialists.
The second part of the title of the work: the need for improvement, is quite limited and shows some shortcomings in the discussed topic. Additionally, the statements: Bladder cancer (BCa) is the tenth most common cancer worldwide, although the in-25 cidence rate is three to four times higher in men than women (line 25-26); when confronted with: On top of this, BCa is one of the most costly to treat overall, and is the most expensive cancer per patient costing between $96,000 to $187,000 (line 111-112) are very shocking and inhumane. I understand that this statement fits perfectly into the thesis of the Authors of the publication "The need for improvement", but it raises moral controversies by putting the weight of human health, life and money on the scales.
A great added value of the work is the Abbreviations section prepared by the Authors.
To sum up, the work will be very interesting if it is prepared more carefully. Performing even a Mini-review on 9 pages of typescript is quite frivolous. I suggest re-researching the literature with particular emphasis on review-type works. Similarly, a review of the literature for a review work, based on only 48 references, does not exhaust the features of a review work. 14 works (29%) cited in references are more than 5 years old. There are very important literature items, but during this time the achievements of world medicine have advanced significantly.
I suggest that the Authors significantly improve their work. This especially applies to the sections: 2. Current treatment options; 3. Limitations with current treatments, 9. Conclusion and References. Sections 4 through 8 require fuller discussion and are somewhat enigmatic in their current form.
Author Response
We would like to thank the reviewer for taking the time to evaluate this manuscript and for their valued comments and feedback.
The review is indeed minimalistic. The Authors of the work presented basic information about current bladder cancer treatment, which is particularly important for both parties - patients and doctors.The methods mentioned in the work are generally known to specialists, but the manner and scope of discussion of currently used bladder cancer treatment methods is significantly limited for non-specialists.
We thank the reviewer for these helpful comments. We intended for this mini review to succinctly reinforce and emphasise the scope of research being undertaken and the potential current and future approaches for this condition. For added clarity, as suggested, we have defined the approaches taken during bladder cancer treatment for all to understand, including for example, TURBT. More description has been added when discussing the current methods on lines 103-105.
The second part of the title of the work: the need for improvement, is quite limited and shows some shortcomings in the discussed topic. Additionally, the statements: Bladder cancer (BCa) is the tenth most common cancer worldwide, although the in-25 cidence rate is three to four times higher in men than women (line 25-26); when confronted with: On top of this, BCa is one of the most costly to treat overall, and is the most expensive cancer per patient costing between $96,000 to $187,000 (line 111-112) are very shocking and inhumane. I understand that this statement fits perfectly into the thesis of the Authors of the publication "The need for improvement", but it raises moral controversies by putting the weight of human health, life and money on the scales.
Thank you for this comment. These facts are indeed shocking but also important and are intended to relay the gravity of the disease and to emphasise the need for new and improved approaches.
A great added value of the work is the Abbreviations section prepared by the Authors.
Thank you for this comment.
To sum up, the work will be very interesting if it is prepared more carefully. Performing even a Mini-review on 9 pages of typescript is quite frivolous. I suggest re-researching the literature with particular emphasis on review-type works. Similarly, a review of the literature for a review work, based on only 48 references, does not exhaust the features of a review work. 14 works (29%) cited in references are more than 5 years old. There are very important literature items, but during this time the achievements of world medicine have advanced significantly.I suggest that the Authors significantly improve their work. This especially applies to the sections: 2. Current treatment options; 3. Limitations with current treatments, 9. Conclusion and References. Sections 4 through 8 require fuller discussion and are somewhat enigmatic in their current form.
I would like to thank the reviewer for their comments. We agree that emphasising some more recent examples is important and have therefore added more recent references and additional revisions, which are highlighted in yellow. These include for instance reference 8, 9, 30 and 47-49.
Reviewer 4 Report
Comments and Suggestions for Authors
The authors present a mini-review on current bladder cancer treatment. The manuscript is well written, well structured, and the images are accurate, with good illustrations.
The manuscript should be proofread by a native speaker, to modify small grammatical errors that are throughout the manuscript.
Comments on the Quality of English LanguageThe manuscript should be proofread by a native speaker, to modify small grammatical errors that are throughout the manuscript.
Author Response
We would like to thank the reviewer for taking the time to evaluate this manuscript and for their valued comments and feedback.
The authors present a mini-review on current bladder cancer treatment. The manuscript is well written, well structured, and the images are accurate, with good illustrations.
The manuscript should be proofread by a native speaker, to modify small grammatical errors that are throughout the manuscript.
I would like to thank the reviewer for their comments. The manuscript has been read by a native speaker to correct any small grammatical mistakes.
Round 2
Reviewer 1 Report
Comments and Suggestions for Authors
Thanks for your revision.
Reviewer 3 Report
Comments and Suggestions for Authors
I would like to thank the Authors for taking into account the comments indicated in the preliminary reviews. This improved the quality of the manuscript.